# Neurodevelopmental Toxicity of Emamectin Benzoate to the Early Life Stage of Zebrafish Larvae (*Danio rerio*)

**DOI:** 10.3390/ijms24043757

**Published:** 2023-02-13

**Authors:** Jie Gu, Liguo Guo, Yuanhui Zhu, Lingling Qian, Lili Shi, Huanchao Zhang, Guixiang Ji

**Affiliations:** 1Key Laboratory of Pesticide Environmental Assessment and Pollution Control, Nanjing Institute of Environmental Sciences, Ministry of Ecology and Environment, Nanjing 210042, China; 2Innovation Center for Sustainable Forestry in Southen China, College of Forestry, Nanjing Forestry University, Nanjing 210037, China; 3Jiangsu Key Laboratory of Preventive and Translational Medicine for Geriatric Diseases, Department of Toxicology, School of Public Health, Medical College of Soochow University, Suzhou 215123, China

**Keywords:** emamectin benzoate, zebrafish, oxidative damage, neurotoxicity

## Abstract

Emamectin benzoate (EMB) is a widely used pesticide and feed additive in agriculture and aquaculture. It easily enters the aquatic environment through various pathways, thus causing adverse effects on aquatic organisms. However, there are no systematic studies regarding the effects of EMB on the developmental neurotoxicity of aquatic organisms. Therefore, the aim of this study was to evaluate the neurotoxic effects and mechanisms of EMB at different concentrations (0.1, 0.25, 0.5, 1, 2, 4 and 8 μg/mL) using zebrafish as a model. The results showed that EMB significantly inhibited the hatching rate, spontaneous movement, body length, and swim bladder development of zebrafish embryos, as well as significantly increased the malformation rate of zebrafish larvae. In addition, EMB adversely affected the axon length of motor neurons in Tg (hb9: eGFP) zebrafish and central nervous system (CNS) neurons in Tg (HuC: eGFP) zebrafish and significantly inhibited the locomotor behavior of zebrafish larvae. Meanwhile, EMB induced oxidative damage and was accompanied by increasing reactive oxygen species in the brains of zebrafish larvae. In addition, gene expression involvement in oxidative stress-related (*cat*, *sod* and *Cu/Zn-sod*), GABA neural pathway-related (*gat1*, *gabra1*, *gad1b*, *abat* and *glsa*), neurodevelopmental-related (*syn2a*, *gfap*, *elavl3*, *shha*, *gap43* and *Nrd*) and swim bladder development-related (*foxa3*, *pbxla*, *mnx1*, *has2* and *elovlla*) genes was significantly affected by EMB exposure. In conclusion, our study shows that exposure to EMB during the early life stages of zebrafish significantly increases oxidative damage and inhibits early central neuronal development, motor neuron axon growth and swim bladder development, ultimately leading to neurobehavioral changes in juvenile zebrafish.

## 1. Introduction

Emamectin benzoate (EMB), a new highly effective semisynthetic biological agent produced from avermectin B1, is widely used in production activities to eliminate agricultural pests [1]. Due to its high efficiency and long period of validity, EMB is widely used in the pest control of vegetables, fruit trees, cotton, and field crops as an alternative to highly toxic pesticides [2,3,4]. In addition, EMB is used as an additive in fish feeds to control parasites, commonly known as sea lice, in aquaculture [5]. Studies have reported a doubling of salmon production in Scotland between 2002 and 2015 and a 5-fold increase in the use of EMB [6]. EMB can enter the marine environment through fish feces or feed and settle into the sediment; its degradation rate is very slow, and it gradually diffuses into the water column over time [2]. Recent studies have reported that EMB was detected in sediments beneath Atlantic fish farms at fairly high levels, with concentrations ranging from 0 to 366 μg/kg (wet weight) [7]. In the vicinity of five aquaculture locations along the Norwegian coast, the concentrations of EMB detected in sediments exceed the UK Environmental Quality Standards [8]. This is likely to cause toxic effects on other aquatic organisms, which in turn affects population dynamics, community structure and food chain function.

EMB is moderately soluble in water (24 mg/L) and can be easily transferred to the aquatic environment [9]. EMB is classified as being very toxic to aquatic organisms and may cause long-term adverse effects in the aquatic environment according to the European Food Safety Authority [10]. The United States Environmental Protection Agency showed that the LC_50_ values of EMB on sheephead minnow (*Cyprinodon variegatus*), rainbow trout (*Oncorhynchus mykiss*), bluegill sunfish (*Lepomis macrochirus*) and fathead minnow (*Pimephales promelas*) were 1.43 mg/L, 0.174 mg/L, 0.18 mg/L and 0.194 mg/L, respectively [11]. Research on EMB residues in surface water is very limited. Benoit et al. reported that the sediment concentration of EMB downstream of freshwater aquaculture facilities ranged from <0.01–2.5 mg/kg [12]. The residue of EMB in the surface water of Chaohu Lake is lower than the detection limit (3.6 μg/L), and the residue in rice field water is 0.54 mg/L [13]. Although the residual concentration is low, it is considered to be persistent in the aquatic environment, with half-lives of more than 1000 days in water or sediment [10]. These characteristics suggest that EMB might cause toxicological effects to aquatic organisms for long periods of time. However, our in-depth understanding of EMB toxicity to aquatic organisms such as fish is limited.

EMB shares the same mechanism of action as avermectin, which mainly acts on the γ-aminobutyric acid (GABA) system of insect neuronal synapses or neuromuscular synapses, thus allowing a large amount of chloride ions to enter the nerve cells, causing loss of cell function and disrupting nerve conduction, leading to the eventual death of insects due to paralysis and food refusal [14]. Previous studies on the neurotoxicity of EMB have focused on rats as a model; for example, black cohosh oil (NSO) is a potential protective and therapeutic agent against EMB-induced neurotoxicity, relying mainly on its antioxidant, anti-inflammatory and anti-apoptotic activities [15]. In 1997, Wise et al. reported that high-dose EMB exposure in rats during gestation and lactation produced evidence of neurotoxicity in F1 offspring and established a no-observed-adverse-effect level (NOAEL) of 0.6 mg/kg/day for EMB on developmental neurotoxicity [16]. With the widespread use of EMB in agriculture, forestry and fishery production, its pollution of the aquatic environment and the threat it poses to aquatic organisms cannot be underestimated. However, the adverse effects of EMB on the nervous system of aquatic animals as well as the underlying mechanisms have not yet been well investigated.

As a nonmammalian model organism, zebrafish (*Danio rerio*) have been widely used in toxicity assessment because of their advantages of small size, low cost, rapid development, and easy observation [17,18]. Furthermore, nervous system development in zebrafish is very similar to that in mammals [19]. Therefore, zebrafish is used as a promising animal model for evaluating the neurodevelopmental toxicity of environmental pollutants [20].

Because of the widespread use of EMB and its high detection in the environment, in this study we used a zebrafish model to investigate the neurotoxic effects of EMB at different concentrations and related mechanisms. The effects of EMB on survival rate, hatching rate, malformation, neurobehavioral functions, nervous development, transcriptional changes in marker genes associated with neurodevelopment and oxidative stress were evaluated. Our findings provide further evidence to evaluate the safety of EMB.

## 2. Results

### 2.1. Developmental Toxicity of EMB to Zebrafish Embryos and Larvae

Zebrafish at the embryonic stage (4 hpf) were exposed to acute toxicity with EMB treatment groups at 0.1, 0.25, 0.5, 1, 2, 4 and 8 μg/mL concentration gradients. As shown in Figure 1a, the significant increase in the mortality of zebrafish in the 0.5, 1, 2, 4 and 8 μg/mL exposure groups compared to the control group showed a dose–effect relationship (Chi-square test: *p* < 0.05) with a 96 h-LC_50_ of 1.176 μg/mL. Since the mortality rate in the 4 and 8 μg/mL exposure groups was close to 100% at 144 hpf, none of the 4 and 8 μg/mL contamination groups were available for subsequent experiments. Similarly, EMB significantly inhibited the zebrafish embryo hatching rate, spontaneous movement, zebrafish body length, and swim bladder area compared to the control group (Chi-square test: *p* < 0.05) and significantly increased the rate of malformations in zebrafish larvae, e.g., spinal curvature (SC), yolk sac edema (Yse), tail deformation, tail malformation (TM), and pericardial edema (PE). Zebrafish deformities were significantly increased in the 0.5, 1 and 2 μg/mL exposure groups compared to the control group. In addition, we also found that EMB had a developmental inhibitory effect on the swim bladder of zebrafish larvae, with 0.5, 1 and 2 μg/mL swim bladder areas being 82.5%, 48.6% and 11.3% of the control group, respectively. In addition, the EC_50_ of EMB for zebrafish hatching rate, spontaneous movement, body length, spinal curvature malformation rate, yolk sac edema malformation rate and swim bladder area were 2.15, 3.54, 21.72, 1.72, 0.94 and 0.95 μg/mL, respectively. The above results suggest that YSE malformation rate is a sensitive indicator.

### 2.2. Effect of EMB on the Behavior of Zebrafish Larvae

Zebrafish embryos were continuously exposed to different concentration groups of EMB until the sixth day, and the movement trajectories of zebrafish larvae under light stimulation were recorded for 40 min in each treatment group using the zebrafish behavior monitoring box. As shown in Figure 2a–d, the movement distance and average speed were significantly inhibited in the exposed groups compared to the control group in a dose-dependent manner (ANOVA test: *p* < 0.05). Meanwhile, the effect of light stimulation disappeared in the 1 and 2 μg/mL-treated zebrafish larvae compared to the control group, which showed a significant dullness (significantly longer dullness time with increasing concentration). The dwell time of zebrafish larvae in the 0.5, 1 and 2 μg/mL treatment groups was significantly prolonged, by 1.29, 1.63 and 1.85 folds, respectively, compared to the control group.

### 2.3. Effect of EMB on Central Nervous Development in Zebrafish Larvae

Based on the inhibition of locomotor behavior of 6 dpf zebrafish larvae by EMB, we used a central nervous transgenic zebrafish line (HuC: eGFP) to investigate the effect of EMB on central nervous development. As shown in Figure 3, EMB treatment at 0.5, 1, and 2 μg/mL significantly inhibited neuronal development in the brain and spinal cord of the zebrafish at 72 and 144 hpf compared to the control (ANOVA test: *p* < 0.05). The above results suggest that EMB has a negative effect on the central neural development of zebrafish larvae.

### 2.4. Effect of EMB on Motor Nerve Development in Zebrafish Larvae

We also used a motor nerve transgenic zebrafish line (HB9: eGFP) to investigate the effect of EMB on motor nerve development in zebrafish larvae. As shown in Figure 4, the axon lengths of motor neurons in the transgenic HB9-GFP zebrafish line treated with 1 and 2 μg/mL EMB at 72 hpf were reduced by 33.0% and 46.2%, respectively, compared to the control group. At 144 hpf, 0.5, 1 and 2 μg/mL EMB treatment significantly reduced the axon lengths of zebrafish motor neurons compared to the control group (ANOVA test: *p* < 0.05). These results suggest that continuous exposure to EMB has a detrimental effect on motor nerve development in zebrafish larvae.

### 2.5. Effect of EMB on the Antioxidant System of Zebrafish

Figure 5 shows the alteration of the antioxidant system in 6 dpf zebrafish larvae after EMB exposure. The 0.5, 1 and 2 μg/mL EMB treatment groups significantly induced the production of reactive oxygen species in the head of zebrafish compared to the control group, 2.4-, 3.5- and 5.1-fold with respect to the control, respectively. In addition, CAT, SOD activity and MDA contents were significantly higher in zebrafish larvae exposed to 0.5, 1 and 2 μg/mL EMB compared to the control (Figure 5c–e) (ANOVA test: *p* < 0.05), and the above results suggest that EMB has a negative effect on the antioxidant system of zebrafish larvae.

### 2.6. Effects of EMB on Oxidative Stress-, Neurodevelopment-, GABA Pathway- and Swim Bladder Development-Related Gene Expression in Zebrafish Larvae

To further explore the neurological and developmental toxicity of EMB at the gene level, the effects of EMB at 144 hpf on genes involved in oxidative stress (*cat*, *sod*, and *Cu/Zn-sod*), GABA neural pathway-related (*gat1*, *gabra1*, *gad1b*, *abat*, and *glsa*), neurodevelopmental-related (*syn2a*, *gfap*, *elavl3*, *shha*, *gap43* and *Nrd*) and swim bladder development-related (*foxa3*, *pbxla*, *mnx1*, *has2* and *elovlla*) genes were evaluated. As shown in Figure 6, EMB reduced the expression of oxidative stress-, neurodevelopment- and swim bladder development-related genes in a dose-dependent manner compared to the control group (ANOVA test: *p* < 0.05). In addition, gat1 gene levels was significantly downregulated at 2 μg/mL compared to the control group; however, gabra1, gad1b and glsa gene levels were dose-dependently upregulated (ANOVA test: *p* < 0.05).

## 3. Discussion

While the use of pesticides has given a great impetus to modern agriculture, it has also caused serious pollution to the aquatic ecosystem, and in addition, it may result in potential long-term harm to aquatic organisms and even human health. EMB is readily accessible to the aquatic environment as a forest pesticide and is also used as a feed additive to control fish parasites, but its toxic effects on aquatic organisms and related mechanisms are still poorly understood.

In the present study, zebrafish embryos were used as an aquatic model to study the neurodevelopmental effects of EMB. In this study, we found that zebrafish showed developmental toxicity after early embryonic exposure to EMB (≥0.5 μg/mL), mainly in the form of increased embryonic mortality and malformation, decreased hatching rate and shortened body length. This is consistent with previous studies in which EMB at different concentrations (0.5, 1.0, 1.5 and 2 μg/mL) caused developmental toxicity in zebrafish [21]. In addition, we observed an inhibitory effect of EMB on swim bladder development in zebrafish larvae. The swim bladder is an important functional organ that helps fish with swim bladders perform physiological activities such as respiration, hydrostatics, and sound sensing. Previously, diazinon exposure was shown to cause a reduced hatching rate and shorter body length in medaka (*Oryzias latipes*) embryos with severe pericardial and yolk sac edema and noninflated swim bladders [22]. Polymyxin exposure caused developmental malformations in zebrafish embryos, including the absence of the hind bladder [23]. This is consistent with our findings, suggesting that some pesticide exposures can affect the normal development or inflation of the swim bladder and are a potential threat to the survival of fish with swim bladders.

Motor behavior assays are used to investigate the effects of environmental factors on biological toxicity, and motor behavior characteristics can partially reflect the neurodevelopment of an organism. Previous studies have shown that exposure of rats to EMB (100 mg/kg administered by gavage daily) resulted in adverse behavioral, motor, and cognitive neurotoxic effects, possibly due to oxidative damage, inflammation, and reduced levels of brain-derived neurotrophic factor (BDNF) [24]. In another study, evaluating the developmental neurotoxicity caused by emamectin benzoate in Sprague–Dawley rats, offspring derived from the high-dose group showed exposure related effects in behavioral tests [16]. In the present study, zebrafish exposed to EMB showed significant inhibition of locomotor behavior (decreased locomotor speed, marked sluggishness and disappearance of light and dark field stimuli), suggesting that EMB may be neurodevelopmentally toxic, which is consistent with the above findings. Additionally, the dysmorphic development of zebrafish caused by EMB can have a negative effect on the locomotor behavior of larval fish. In addition, to further confirm the effect of EMB on neurodevelopment, we next used neurotransgenic zebrafish to assess the neurotoxicity of EMB.

Altered early motor behavior in zebrafish larvae is often associated with their neurodevelopment, so we used central and motor nerve transgenic zebrafish (HuC: eGFP and hb9: eGFP) to assess the effect of EMB on neurodevelopment in zebrafish larvae at different concentrations. In Tg (HuC: eGFP) zebrafish, GFP expression is driven by the promoter of the neurodevelopmental marker gene *elavl3*, which specifically expresses the marker gene in neurons throughout the body. The *elavl3* gene, alias huc, was first identified in *Drosophila*, and deletion of this gene causes visual neurological defects accompanied by neurological dysplasia and severe lethality [25]. In Tg (hb9: eGFP) zebrafish larvae, the expression of GFP was driven using the key gene for motor neurodevelopment, hb9, to label motor neurons throughout the zebrafish body [26]. Previous studies investigating the neurotoxic effects of nanotitanium dioxide using both transgenic zebrafish showed that nanotitanium dioxide treatment adversely affected motor neuron axon length in Tg (hb9: eGFP) zebrafish and central nervous system (CNS) neurons in Tg (HuC: eGFP) zebrafish and significantly inhibited locomotor behavior in zebrafish [27]. Similarly, BPAF, a typical analog of BPA, significantly reduced motor neuron length and decreased CNS neuron development in the transgenic line hb9-GFP zebrafish [28]. These studies are consistent with our results that different concentrations of EMB treatment inhibited central developmental neurons and motor neurons to different degrees at 72 and 144 hpf, respectively, while the dose–effect relationship indicated that EMB inhibition of locomotor behavior in zebrafish was positively correlated with central and motor nerve damage.

Regarding the potential mechanisms of EMB-induced toxicity, several studies have shown that oxidative stress plays a key role in the mechanism of EMB-induced cardiotoxicity [21]. Oxidative stress is an important toxicity mechanism for pesticide toxicity. For example, a previous study showed that the organophosphorus insecticide propamocarb (PFF) induced oxidative stress and induced DNA damage in the freshwater snail *Lymnea luteola* [29]. In addition, it has been shown that Nigella sativa oil (NSO) has antioxidant properties that protect against neurotoxicity induced by emamectin benzoate, suggesting that EMB has oxidative damage capacity [15]. In addition, Özge Temiz used a male mouse model by oral administration of EMB at doses of 25, 50 and 100 (mg/kg/day) for 14 days, and the in vivo results suggested that EMB significantly induced oxidative damage in liver tissues, as evidenced by increased TBARS levels and decreased GSH levels [30].

Therefore, in this study, we sought to investigate whether oxidative stress plays a role in EMB-mediated neurodevelopmental toxicity induced by zebrafish larvae. Our study revealed that the antioxidant enzymes CAT and SOD and the lipid peroxidation byproduct MDA were all significantly upregulated after EMB exposure, and correspondingly, the reactive oxygen species levels in the heads of zebrafish larvae were significantly increased. In addition, the expression of cat, sod and Cu/Zn-sod mRNAs at the gene level also received significant inhibition by EMB. In a previous study on the neurotoxicity of pollutants (BPF) on zebrafish larvae through oxidative damage, CAT and SOD enzyme activities of zebrafish larvae were significantly elevated at 3 dpf and significantly inhibited by 6 dpf, but cat and sod1 were significantly inhibited at the gene level at both time periods [31], which is partially consistent with our results. In our study, zebrafish larvae showed an increase in MDA content under EMB, which is consistent with previous studies [32]. The above results suggest that EMB-induced neurotoxicity may be due to enhanced oxidative stress, which leads to a significant increase in lipid peroxidation and reactive oxygen species. Of course, if we want to further verify that oxidative damage is an important pathway for the neurotoxicity of EMB for zebrafish larvae, we should also use antioxidants to perform protective tests.

To further explore the molecular mechanisms of EMB-induced nerve damage in zebrafish, the expression of genes involved in oxidative stress-related, GABA receptor pathway-related, neurodevelopment-related and swim bladder development-related genes was examined. Numerous studies have shown that abnormalities in GABA and its receptors are associated with neurological disorders such as epilepsy, depression, anxiety, and amnesia [33]. The *gat1*, *gabra1*, *gad1b*, *abat*, and *glsa* genes play important roles in the exertion of major GABA physiological activities, such as the regulation of GABA neurotransmitter synthesis, conversion, and inactivation, respectively [34,35]. In this study, the *gabra1*, *gad1b* and *glsa* genes were significantly upregulated after EMB exposure, suggesting that EMB promotes the transport and conversion of GABA, which causes neurotransmitter disruption. The genes *syn2a*, *gfap*, *elavl3*, *shha*, *gap43* and *Nrd* play key regulatory roles in neuronal development, transmitter release and synapse formation, respectively [36]. It has been previously shown that exogenous contaminants cause neurotoxicity in zebrafish by altering the expression of genes related to neurodevelopment. For example, bisphenol substitutes (BPS and BPF) inhibited the expression of neurodevelopmental genes such as *a1-tubulin*, *elavl3*, *gap43*, *mbp*, *syn2a*, and *gfap*, leading to sluggish movements in zebrafish larvae [31,37], which is consistent with our findings. EMB has a negative effect on the development and differentiation of neurons in zebrafish larvae. Furthermore, the downregulation of *elavl3* mRNA transcript levels is consistent with reduced green fluorescence in HuC-GFP transgenic zebrafish. Finally, we examined the expression of important genes regulating swim bladder development in response to morphological changes in swim bladder development after EMB exposure. The *foxA3* is normally expressed in swim bladder buds and other organs of intestinal origin [38]. The *pbx1* is an important regulator of swim bladder progenitor development and plays an important regulatory role in the late development of the swim bladder [39]. The *mnx1*, *has2* and *elovl1a* genes are considered marker genes for the development of the epithelium, mesoderm and mesocortex of the swim bladder, respectively [40,41]. The downregulation of these genes in the EMB-treated group suggests that EMB may impair the development of the swim bladder from the epithelial emergence stage to the growth stage, resulting in an abnormal morphology of the swim bladder.

## 4. Materials and Methods

### 4.1. Chemicals and Reagents

Emamectin benzoate (EMB, CAS No. 155569-91-8, purity > 95%) was purchased from Jiangxi Ruiweier Biotechnology Company (Ji’an, China). The Catalase (CAT) detection kit, Total Superoxide Dismutase (SOD) activity detection kit, Lipid Peroxidation (MDA) Assay Kit, Enhanced BCA Protein Assay Kit, and reactive oxygen species (ROS) detection kit were purchased from Beyotime Company (Shanghai, China).

### 4.2. Zebrafish Maintenance and Embryo Collection

The following zebrafish strains were used in this study: common wild-type zebrafish (AB), transgenic zebrafish Tg (HuC: eGFP) specifically labeled with central nerve, and transgenic zebrafish Tg (hb9: eGFP) specifically labeled with motor nerve, all purchased from Wuhan Institute of Aquatic Biology, Chinese Academy of Sciences. The zebrafish were reared in a recirculating fish culture system at the Nanjing Institute of Ecological and Environmental Sciences, Ministry of Ecology and Environment. The water used for zebrafish culture was aerated tap water with water temperature controlled at 27.5~28.5 °C, dissolved oxygen greater than 6 mg/L, pH maintained between 6.5~7.5, salinity of 0.25~0.50‰ and conductivity of 500~800 μS/cm. The light time of the zebrafish culture room was 14 h:10 h (day:night). Adult zebrafish were fed twice per day, and excess feed and excrement were removed in a timely manner. One-third water of the breeding system was replaced by freshly prepared aeration water every morning.

On the night before breeding, male and female fish are placed in the breeding box at a ratio of 1:1 or 2:1 according to the embryo demand. Early the next morning, the partition in the breeding box was removed, and the male fish started to chase the female fish to spawn under the stimulation of light. After spawning, the male and female breeding fish were separated and reared in their original tanks, and the fertilized eggs were collected. The normally developed fertilized eggs were selected under a microscope, the dead embryos were picked out after 4~5 h of incubation, and the remaining well-developed embryos were used for subsequent testing of toxicity. All animal experiments were conducted in accordance with the guidelines for the care and use of laboratory animals of Nanjing Institute of Environmental Science.

### 4.3. Embryonic Waterborne Exposure

Acute toxicological studies of EMB were performed on zebrafish embryos according to the previous study [37]. Briefly, ten zebrafish embryos were cultured in 6-well plates exposed to 5 mL of EMB (0.1, 0.25, 0.5, 1, 2, 4 and 8 μg/mL) treatments for 144 h. The EMB solution was renewed daily to ensure the stability of the solution concentration. During the experiment, the effects of different concentrations of EMB exposure on the hatching rate, mortality and malformation rate of zebrafish embryos were photographed and recorded twice daily using a stereomicroscope (Nikon, SMZ25, Tokyo, Japan), while dead embryos were removed as soon as possible to prevent contamination (the percentage of surviving, hatching and deformed fish divided by the total fish, *n* = 3). At 24 hpf, spontaneous movement (times, *n* = 12) of zebrafish embryos in different treatment groups was measured with EthoVision^®^ XT 16 (Noldus, Wageningen, The Netherlands). At the same time, at 144 hpf, zebrafish larvae from different treatment groups were anesthetized with 0.02% MS-222, and the body length (μm, *n* = 12) and swim bladder area (μm^2^, *n* = 12) of the larvae were measured using NIS-Elements D software 5.41.00 (Nikon, Tokyo, Japan).

### 4.4. Detection of Locomotor Behavior of Zebrafish Larvae

Zebrafish larvae were randomly selected for continuous exposure to different concentrations of EMB treatment groups for 6 days and placed in 24-well plates with one fish and 2 mL of clean water per well (*n* = 24). The zebrafish larval behavioral trajectory tracking system (Noldus, Wageningen, The Netherlands) was turned on, and the observation area and detection program were set up according to the reference [27] and accompanied by photoperiodic light and dark field stimulation (10 min alternating light and dark). The movement trajectories of each group of zebrafish larvae within 40 min were collected separately using EthoVision^®^ XT 16 (Noldus, Wageningen, The Netherlands), and the distance of movement behavior, procession time and the behavior trajectory of swimming were derived using the software. Then, the total distance travelled, the dwell time and the average speed under different light conditions were calculated for each group of fish and included in the statistics separately.

### 4.5. Fluorescence Image Observation of Neural Transgenic Zebrafish

Based on the effect of EMB on the locomotor behavior of zebrafish, zebrafish embryos of transgenic line HuC-GFP and transgenic line hb9-GFP were next exposed to different concentrations of EMB solution, which was used to assess the effect of EMB on the development of the nervous system of zebrafish larvae. Following the methodology of previous studies [28], we selected 72 and 144 hpf (early neurodevelopmental stage) to observe the expression of central and motor nerves in zebrafish larvae. Zebrafish larvae (*n* = 12) were randomly selected from each treatment group and fixed in 4% paraformaldehyde for 30 min. Images of the CNS and motor nerves of transgenic line HuC-GFP and transgenic line hb9-GFP larvae were taken and recorded by a stereo fluorescence microscope (Nikon, SMZ25, Tokyo, Japan). The fluorescence intensity of the corresponding green fluorescent protein (HuC: eGFP) and axon length of the motor nerve (hb9: eGFP) in zebrafish larvae were quantified using NIS-Elements D software (Nikon, Tokyo, Japan).

### 4.6. Detection of Oxidative Stress Levels

Fertilized embryos were treated with different concentrations of EMB for 6 days. A random selection of 6 dpf zebrafish larvae from each treatment group (20 larvae pooled as one sample, *n* = 3) was homogenized for analysis of SOD, CAT and MDA. After calibration of the concentrations using the Enhanced BCA Protein Assay Kit, SOD, CAT activities and MDA content were assayed as described in the kit instructions.

According to the previous method, in brief, 12 live larvae in each treatment group were randomly selected and stained with the diluted fluorescent probe 2′,7′-dichlorodihydrofluorescein diacetate (DCFH-DA) at 28 °C for 20 min [42]. After washing three times with PBS, the fluorescence intensity was photographed and measured with a stereo fluorescence microscope (Nikon, SMZ25, Tokyo, Japan). All experiments were performed in three biological replicates.

### 4.7. Real-Time PCR

Total RNA was extracted from zebrafish larvae (20 larvae pooled as one sample, *n* = 3) of different treatment groups using TRIzol reagent (Invitrogen, Carlsbad, CA, USA) according to the manufacturer’s instructions, and its concentration was determined. RNA was reverse transcribed to cDNA using PrimeScript^®^ RT (TaKaRa, Tokyo, Japan). The housekeeping gene β-actin was used as an internal reference. The expression of this gene was normalized to that of the housekeeping gene by using the 2^−ΔΔCt^ method [43]. Based on the results of EMB neurotoxicity and developmental toxicity in juvenile zebrafish, we selected oxidative stress (*cat*, *sod*, and *Cu/Zn-Sod*), GABA neural pathway-related (*gatl*, *gabral*, *gadlb*, *abat*, and *glsa*), neurodevelopmental-related (*syn2a*, *gfap*, *elavl3*, *shha*, *gap43*, and *Nrd*), and swim bladder development-related (*foxa3*, *pbxla*, *mnx1*, *has2* and *elovlla*) genes for the next step. Each sample was taken in triplicate, and the primer sequences of the selected genes are listed in Appendix A.

### 4.8. Statistical Analysis

All the contentious data were expressed as the mean ± SEM. One-way ANOVA was used for evaluating variance between groups, and the difference between the control group and each different exposure group was evaluated by Dunnett’s test. All statistical analyses were performed using SPSS (version 20.0, Chicago, IL, USA). *p* < 0.05 was identified as statistically significant.

## 5. Conclusions

In summary, in this study, EMB was shown for the first time to inhibit early central neuronal development, motor neuron axon growth and swim bladder formation in zebrafish, inducing neurodevelopmental toxicity in zebrafish larvae, which ultimately leads to behavioral changes. This study novelly uses two neurotransgenic zebrafish for rapid screening of EMB neurotoxicity, which facilitates later rapid screening of neurotoxicity against a large number of contaminants. These data strengthen our understanding of the mechanisms of EMB toxicity and may contribute to ecological risk assessment; however, further studies are still needed to explore the signaling pathways involved in EMB developmental neurotoxicity as well as swim bladder development.

## Figures and Tables

**Figure 1 ijms-24-03757-f001:**
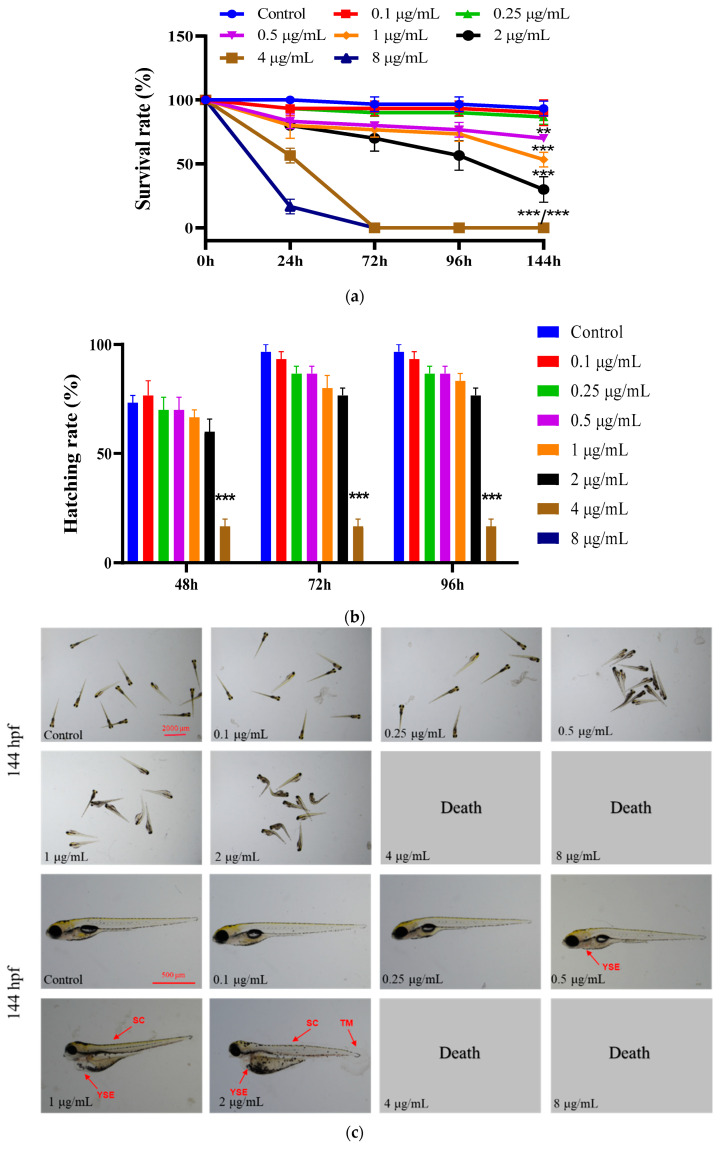
Developmental toxicity of EMB exposure to zebrafish embryos and larvae. (**a**) Survival rate—percentage of surviving fish divided by the total number of fish, *n* = 3; (**b**) hatching rate, percentage of hatching fish divided by the total number of fish, *n* = 3; (**c**) representative picture; 24 hpf (**d**) spontaneous movement, *n* = 12; (**e**) 144 hpf body length, *n* = 12; (**f**) abnormal spinal curvature rate, percentage of deformed fish divided by the total number of fish, *n* = 3; (**g**) abnormal yolk sac edema rate, percentage of deformed fish divided by the total number of fish, *n* = 3; and (**h**) results of fish bladder area statistics, *n* = 12. * *p* < 0.05, ** *p* < 0.01, *** *p* < 0.001, compared with control groups.

**Figure 2 ijms-24-03757-f002:**
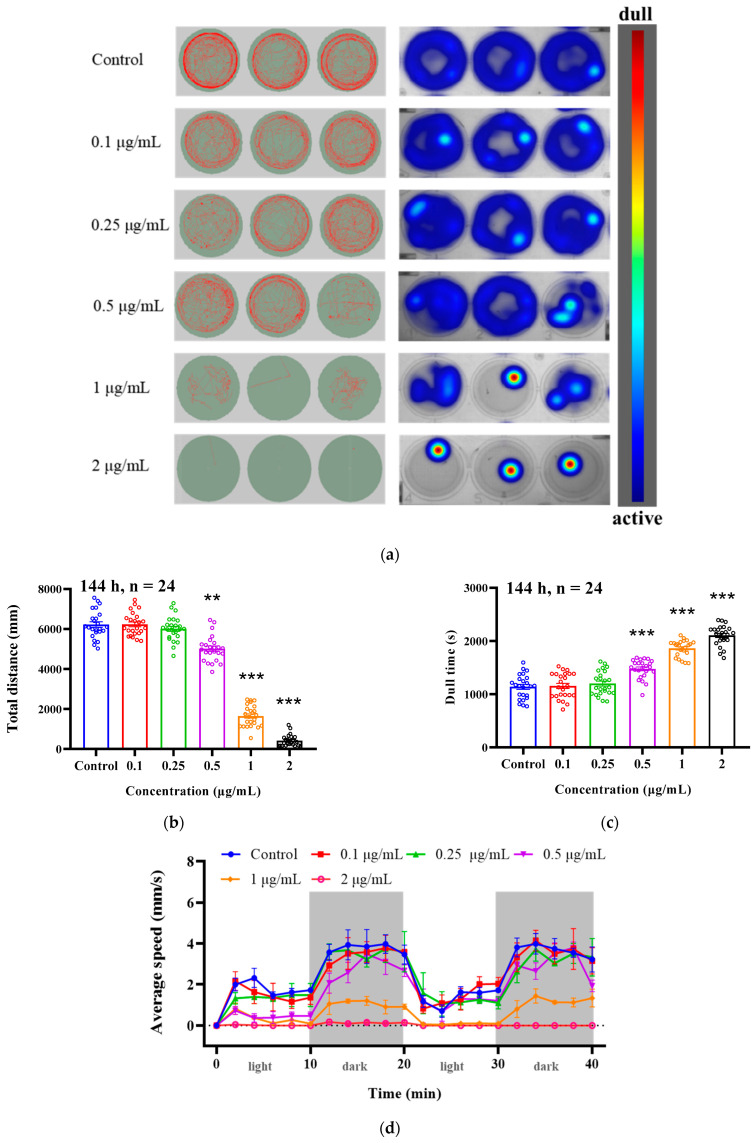
Altered locomotor behavior induced by exposure to EMB. Representative motion trajectories and heatmaps of motion trajectories (**a**), total distance of movement (**b**), dwell time (**c**) and mean velocity under bright- and dark-field stimulation (**d**). (*n* = 24 in each group) ** *p* < 0.01, *** *p* < 0.001, compared to the control group.

**Figure 3 ijms-24-03757-f003:**
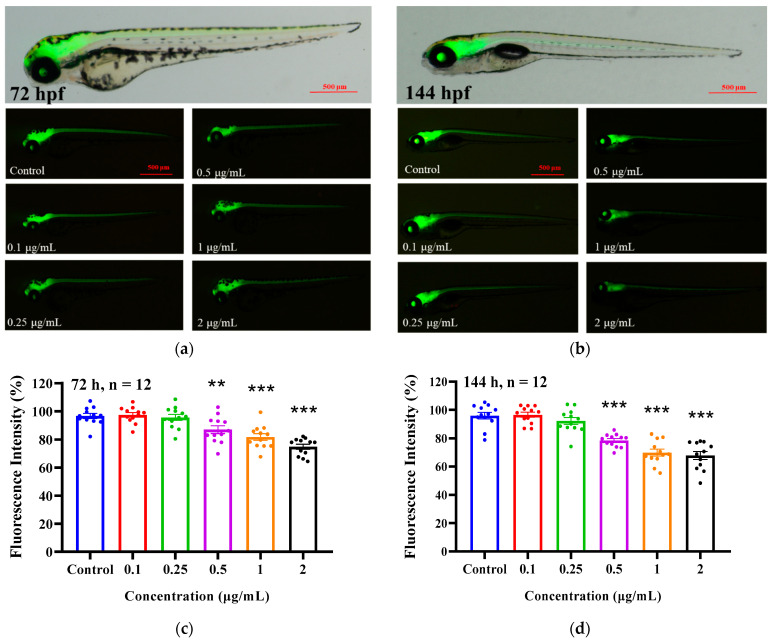
EMB exposure suppressed the expression of central nervous fluorescence in Tg (HuC-GFP) zebrafish. Green fluorescence images (**a**,**b**) and fluorescence statistics (**c**,**d**) of central nervous system (CNS) neurogenesis in Tg (HuC-GFP) zebrafish at 72 and 144 hpf (*n* = 12 in each group). ** *p* < 0.01, *** *p* < 0.001, compared to the control group.

**Figure 4 ijms-24-03757-f004:**
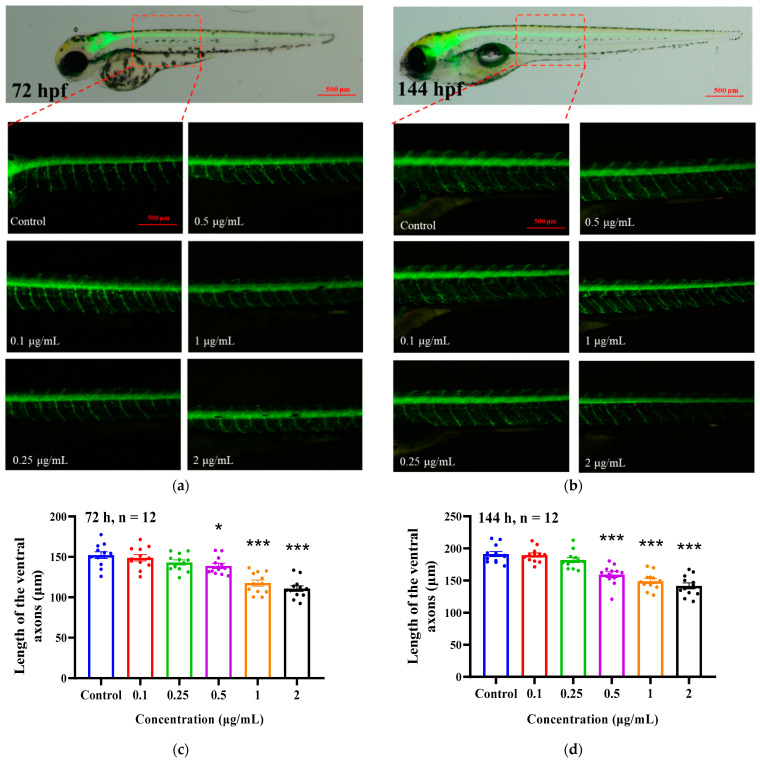
EMB exposure negatively affects motor neuron axon length in Tg (hb9-GFP) zebrafish. Green fluorescence images of motor neuron axon length in Tg (hb9-GFP) zebrafish at 72 and 144 hpf (**a**,**b**) and fluorescence statistics (**c**,**d**) (*n* = 12 in each group). * *p* < 0.05, *** *p* < 0.001, compared with control groups.

**Figure 5 ijms-24-03757-f005:**
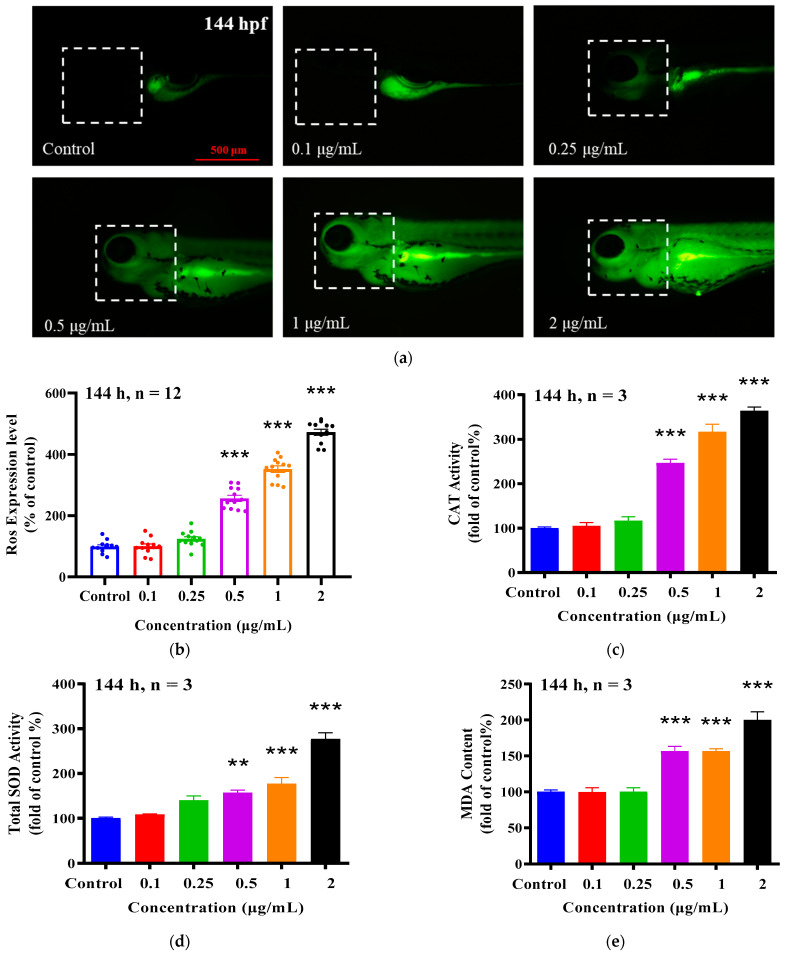
EMB exposure induced oxidative damage in zebrafish. At 144 hpf, representative images of reactive oxygen species in the head of zebrafish larvae (**a**), fluorescence statistics (**b**), CAT enzyme activity (**c**), SOD enzyme activity (**d**), and MDA content (**e**), (20 larvae pooled as one sample, *n* = 3). ** *p* < 0.01, *** *p* < 0.001, compared with control groups.

**Figure 6 ijms-24-03757-f006:**
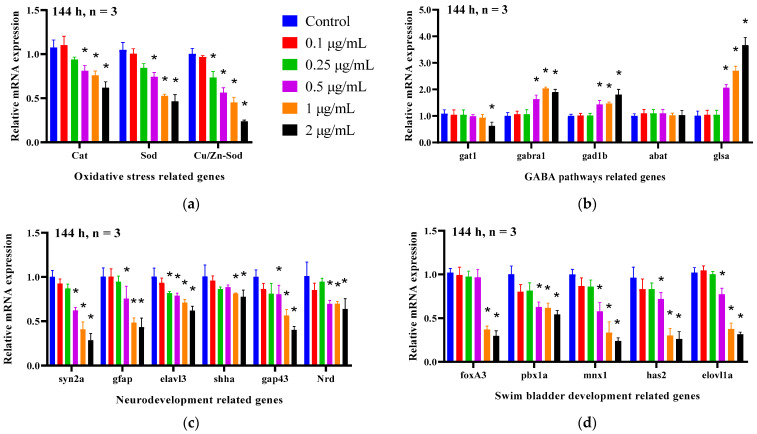
Gene expression changes induced by exposure to EMB. EMB exposure altered oxidative stress (**a**), the GABA pathway (**b**), neurodevelopment (**c**), and swim bladder development-related genes (**d**), (20 larvae pooled as one sample, *n* = 3). * *p* < 0.05, compared with the control groups.

## Data Availability

Not applicable.

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
