# Peer review of "Neurodevelopmental Toxicity of Emamectin Benzoate to the Early Life Stage of Zebrafish Larvae (*Danio rerio*)"

_ijms, 2023, doi:10.3390/ijms24043757_

Round 1
Reviewer 1 Report
1. This paper requires extensive language and formatting improvements.
2. what is the suggestion of this study for future works?
3. Objective should be elaborated on in the introduction section.
4. Please compare your results with previous works.
5. Concept of study is diffused. Need to explain and correlate of obtained results
6. Need to rewrite the results in detail.
7. The discussion section needs a few improvements. Currently, it is diffuse and provides no real insights.
8. References need to be rechecked and formatted as per the journal's guidelines.
9. Please add these references for your discussion part of the manuscript and bold your study novelty :
Investigation of biological accumulation and eco-genotoxicity of bismuth oxide nanoparticle in fresh water snail Lymnaea luteola https://doi.org/10.1016/j.jksus.2021.101355
Detection of oxidative stress and DNA damage in freshwater snail Lymnea leuteola exposed to profenofos
https://doi.org/10.1007/s11783-018-1039-6
Genotoxicity in the freshwater gastropod Lymnaea luteola L: assessment of cell type sensitivities to lead nitrate
http://dx.doi.org/10.1080/02757540.2016.12755873
Author Response
Revise response
Dear Editors and Reviewers,
We highly appreciate the valuable comments of referees on our manuscript “Neurodevelopmental toxicity of emamectin benzoate to the early life stage of zebrafish larvae (Danio rerio)”. The suggestions are quite helpful for us, and these comments will greatly enhance the quality of this manuscript. We have modified the manuscript accordingly, and the detailed corrections are listed below point by point.
We deeply hope that our responses are satisfactory. If you have further questions, please let us know by E-mail. Thank you again for your consideration and we await a favorable response to the revision.
Thanks and Best Regards!
Yours Sincerely,
Guixiang ji
Response to Reviewer 1 Comments
Reviewer #1 Major comments
- This paper requires extensive language and formatting improvements.
Response 1: We are really grateful to you for your helpful comments. We have commissioned Elsevier to do the language touch-up work before submitting the manuscript, and the format we revised as required. In addition, we invited Professor Wang Jun from Nanjing Medical University to revise the language of the article.
- what is the suggestion of this study for future works?
Response 2: Thanks again for your suggestion. The current paper focuses on the screened neurotoxic effects of EMB using acute toxicity exposure in zebrafish embryos. Our study mainly found significant neurodevelopmental toxicity in zebrafish at an exposure concentration of 5μg/ml. Based on the above experimental results, we also carried out long-term chronic effects of EMB on zebrafish at ambient concentrations.
- Objective should be elaborated on in the introduction section.
Response 3: Thanks for your comments. This sentence has been changed as required. “Because of the widespread use of EMB and it’s high detection in the environment, in this study we used a zebrafish model to investigate the neurotoxic effects of EMB at different concentrations and related mechanisms.”
- Please compare your results with previous works.
Response 4: Thanks for your comments. Related studies have been added to our discussion section.
- Concept of study is diffused. Need to explain and correlate of obtained results
Response 5: Thanks for your comments. Our research is focused on a systematic study about the neurotoxic effects of EMB on the development of aquatic organisms. In this study, we used zebrafish as a model to assess the neurotoxic effects and mechanisms of different concentrations (0.1, 0.25, 0.5, 1, 2, 4 and 8 μg/mL) of EMB. We found that EMB exposure produced significant toxic phenotypes in zebrafish, including changes in body length, swim bladder development, neurodevelopment and behavior, suggesting that EMB is neurodevelopmentally toxic. In addition, the study initially explored the mechanism of toxic effects of EMB in terms of oxidative stress and gene expression, and found that genes related to the induction of oxidative stress and neurodevelopment were altered. The summary figure of this paper is as follows, which should explain the findings of this paper more clearly.
- Need to rewrite the results in detail.
Response 6: I am very grateful to you for your careful comments, which have been revised as required. We have revised and detailed the results section of 2.1 and 2.2.
- The discussion section needs a few improvements. Currently, it is diffuse and provides no real insights.
Response 7: I am very grateful to you for your careful comments, which have been revised as required.
- References need to be rechecked and formatted as per the journal's guidelines.
Response 8: I am very grateful to you for your careful comments. We have revised the format of the references in accordance with the requirements of the journal.
- Please add these references for your discussion part of the manuscript and bold your study novelty.
Investigation of biological accumulation and eco-genotoxicity of bismuth oxide nanoparticle in fresh water snail Lymnaea luteola https://doi.org/10.1016/j.jksus.2021.101355
Detection of oxidative stress and DNA damage in freshwater snail Lymnea leuteola exposed to profenofos
https://doi.org/10.1007/s11783-018-1039-6
Genotoxicity in the freshwater gastropod Lymnaea luteola L: assessment of cell type sensitivities to lead nitrate
http://dx.doi.org/10.1080/02757540.2016.12755873
Response 9: Thanks to your comments, I have revised the novelty of this study. We cite the second reference you recommended for the study of oxidative stress in pesticides.

Reviewer 2 Report
In this manuscript, the authors studied the effects of EMB on the developmental neurotoxicity of zebrafish larvae. The following problems are suggested to be improved.
1. The author has studied many aspects of the toxic effects of EMB on zebrafish larvae. In the abstract and the last paragraph of the introduction, the author said that the relevant mechanism of the effect of EMB on neurotoxicity was studied, but the mechanism was not clearly described from the perspective of the article as a whole.
2. The content of Line189-194 repeats the content of the introduction, there is no need to repeat it again.
3. Lines 313-315, the culture condition "water temperature controlled at 27.5~28.5°C, dissolved oxygen greater than 6 mg/L, pH maintained between 6.5~7.5, salinity of 0.25~0.50 ‰ and conductivity of 500~800 μS/cm. ", why are temperature, dissolved oxygen, pH, salinity and conductivity all within a certain range? For a culture system, the training conditions should be the same. For example, the salinity is 0.15-0.5, and the change rate has been 2 times. Please explain why.
4. Line 101, 96h-Lc50 of 1.176ug/ml, how is this calculated? The full paper does not show the dose-effect relationship curve between any index and EMB concentration.
5. How to better connect the impact on many indicators is very critical, and this aspect needs further improvement.
Author Response
Revise response
Dear Editors and Reviewers,
We highly appreciate the valuable comments of referees on our manuscript “Neurodevelopmental toxicity of emamectin benzoate to the early life stage of zebrafish larvae (Danio rerio)”. The suggestions are quite helpful for us, and these comments will greatly enhance the quality of this manuscript. We have modified the manuscript accordingly, and the detailed corrections are listed below point by point.
We deeply hope that our responses are satisfactory. If you have further questions, please let us know by E-mail. Thank you again for your consideration and we await a favorable response to the revision.
Thanks and Best Regards!
Yours Sincerely,
Guixiang ji
Response to Reviewer 2 Comments
Reviewer #2 Major comments
In this manuscript, the authors studied the effects of EMB on the developmental neurotoxicity of zebrafish larvae. The following problems are suggested to be improved.
Answer: Thank you for your affirmation of our article. Your valuable advices will greatly enhance the quality of this manuscript
- The author has studied many aspects of the toxic effects of EMB on zebrafish larvae. In the abstract and the last paragraph of the introduction, the author said that the relevant mechanism of the effect of EMB on neurotoxicity was studied, but the mechanism was not clearly described from the perspective of the article as a whole.
Response 1: Thanks for your comments. Our research is focused on a systematic study about the neurotoxic effects of EMB on the development of aquatic organisms. In this study, we used zebrafish as a model to assess the neurotoxic effects and mechanisms of different concentrations (0.1, 0.25, 0.5, 1, 2, 4 and 8 μg/mL) of EMB. We found that EMB exposure produced significant toxic phenotypes in zebrafish, including changes in body length, swim bladder development, neurodevelopment and behavior, suggesting that EMB is neurodevelopmentally toxic. In addition, the study initially explored the mechanism of toxic effects of EMB in terms of oxidative stress and gene expression, and found that genes related to the induction of oxidative stress and neurodevelopment were altered. The summary figure of this paper is as follows, which should explain the findings of this paper more clearly. In addition, we have revised the Abstract section.
- The content of Line189-194 repeats the content of the introduction, there is no need to repeat it again.
Response2: Thank you for your comments, we have deleted this part of the repeat content.
- Lines 313-315, the culture condition "water temperature controlled at 27.5~28.5°C, dissolved oxygen greater than 6 mg/L, pH maintained between 6.5~7.5, salinity of 0.25~0.50 ‰ and conductivity of 500~800 μS/cm. ", why are temperature, dissolved oxygen, pH, salinity and conductivity all within a certain range? For a culture system, the training conditions should be the same. For example, the salinity is 0.15-0.5, and the change rate has been 2 times. Please explain why.
Response 3: Thanks a lot for your suggestions. These parameter ranges are mainly set according to the optimum culture conditions for zebrafish, and most of the parameters for fish creatures are a range.
- Line 101, 96h-Lc50 of 1.176ug/ml, how is this calculated? The full paper does not show the dose-effect relationship curve between any index and EMB concentration.
Response 4: Thank you for your comments. Use the probabilistic unit regression method to calculate LC50, set the variables in SPSS as concentration, number of deaths, and total and enter the corresponding values, use probabilistic analysis, set the response frequency as mortality, the summary of observations as total, the covariate as concentration, select the log base as 10 for the transformation, select logit for the model, and select the concentration value corresponding to the probability of 0.5 in the pop-up output interface, which is LC50.
Regarding the dose-effect relationship curve which is a good suggestion, according to the results our maximum no-effect dose is 0.25 μg/ml. In addition, the EC50 of EMB for zebrafish hatching rate, spontaneous movement, body length, spinal curvature malformation rate, yolk sac edema malformation rate and swim bladder area were 1.96, 3.54, 21.72, 1.72, 0.94 and 0.95 μg/ml, respectively. The above results suggest that YSE malformation rate is a sensitive indicator.
- How to better connect the impact on many indicators is very critical, and this aspect needs further improvement.
Response 5: Thank you for your comments. We used the zebrafish model to study the neurotoxicity of EMB and its mechanisms, and the transgenic zebrafish to rapidly assess the effects of EMB on neurodevelopment. In addition, the next step was to examine the altered oxidation levels and gene levels in zebrafish to jointly explain the adverse effects of EMB on aquatic organisms. We added the EC50 data of the indicators in the results section for better screening of sensitive indicators.

Round 2
Reviewer 1 Report
English language and style are fine/minor spell check required